# Barriers Facing Direct Support Professionals When Supporting Older Adults Presenting with Intellectual Disabilities and Unusual Dementia-Related Behavior: A Multi-Site, Multi-Methods Study

Karsten Ebbing [1], Armin von Gunten [1], Vincent Guinchat [1], Dan Georgescu [2], Taree Bersier [1], Djamel Moad [1] and Henk Verloo [1,3,*]

1 Department of Psychiatry, Service of Old Age Psychiatry, Lausanne University Hospital, 1008 Prilly, Switzerland
2 Department of Consultation-Liaison Psychiatry, Old Age Psychiatry and Neuropsychiatry, Aargau Psychiatric Services, 5210 Windisch, Switzerland
3 Department of Nursing Sciences, HES-SO University of Applied Sciences and Arts Western Switzerland, 1950 Sion, Switzerland
* Correspondence: henk.verloo@hevs.ch or henk.verloo@chuv.ch

**Abstract:** Introduction: Increased life expectancy among people with intellectual disabilities (ID) raises the risk of their diagnosis being superimposed by behavioral and psychological symptoms of dementia (BPSD). The difficulties facing direct support professionals dealing with this is an emerging, under-investigated issue. The study investigates direct support professionals' perceptions and experiences of their daily support for aging people with ID presenting with superimposed BPSD. Method: Twenty-four direct support professionals from long-term care facilities responded to clinical vignettes and attended focus groups conducted to investigate perceptions and lived experiences of the barriers and struggles they faced. Results: Direct support professionals' reactions to vignettes revealed their difficulties recognizing BPSD superimposed on the known challenging behaviors of people with ID. Focus groups highlighted daily struggles with BPSD, the lack of knowledge about detecting and dealing with them, and associated somatic and psychopathological diseases of aging. Conclusion: Improved knowledge transfer about good practices for person-centered support to aging people with ID presenting with BPSD is strongly recommended.

**Keywords:** disability; aging; BPSD; dementia; specialized educator

## 1. Introduction

In recent decades, improved health, social, and medical services have enabled people with intellectual disabilities (ID) to live longer; however, they thus become exposed to health conditions associated with aging [1–3]. The normal adverse effects of aging are grafted onto pre-existing physical problems and intellectual disability (ID) [4]. The number of people with an ID living into old age and developing dementia continues to increase, and their progressive deterioration presents a wide range of challenges to direct support professionals [5]. The overall life expectancy among people with ID has been estimated at 71.5 years for women and 66.7 years for men. Depending on their level of disability, life expectancy for people with mild ID is 74 years, with moderate ID is 68 years, and with severe ID is 59 years [6–8]. The life expectancy for people with Down syndrome—a subgroup of people with ID—has increased but not yet achieved longevity comparable to other people with ID and is near 60 years [9–12]. The American Association of Intellectual and Developmental Disabilities (AAIDD) defines intellectual and developmental disability as a significant limitation in intellectual functioning and adaptive behavior, which covers many everyday social and practical skills; it originates before the age of 22 [13]. More

recent definitions of ID suggest that the patient's degree of disability—mild, moderate, severe, or profound/complete—should be determined using adaptive functioning rather than the intelligence quotient. The intelligence quotient is, nevertheless, still often a determining factor in daily practice and in the organization of care and support in long-term care facilities [14]. The International Classification of Functioning, Disability, and Health Impairment (ICF) conceptualizes disability as a 'dynamic interaction' between a person's health status, environmental factors, and personal factors [14].

People with ID present higher incidences of heart disease, obesity, osteoporosis, endocrine disorders, sensory disorders, and psychiatric disorders (including early onset dementia) than does the general population [4,15]. The effects of aging bring a further decline in the functional abilities already impaired by ID, increasing the occurrence of age-related, degenerative, and metabolic diseases, a process which may aggravate existing functional impairments or cause new ones [3,16]. Besides a higher incidence of age-related diseases than among subjects without ID, research has highlighted changing clinical manifestations of ID in the context of this new stage in life, although the expression of these impairments will vary significantly among aging people with ID [1,4].

Behavioral changes (BPSD) and functional decline compared to a person's previous functional abilities are the first signs and symptoms of superimposed dementia [17–19]. BPSD can be defined as a heterogeneous range of psychological reactions, psychiatric symptoms, and behaviors resulting from the presence of dementia. It is associated with reduced quality of life, increased risk of mortality, accelerated cognitive decline, and earlier institutionalization for patients, as well as severe burdens for caregivers and relatives and increased financial costs [20]. The diagnostic criteria for dementia among people with ID are the same as those used for the general population. However, the onset of dementia-related behavioral disorders in people with ID is often challenging to establish because of the lack of relevant retrospective or historical clinical data [21–25]. Although dementia is defined as a newly acquired deficit in at least two cognitive functions, with an adverse impact on the activities of daily living [1,26], the present study considered dementia as it was noted in the medical records of people with ID. These records mentioned the cognitive and functional decline observed and documented by direct support professionals and family members, as well as the BPSD involving new unfamiliar behaviors different from the known challenging behaviors due to ID. We will use unfamiliar behavior or unusual behavior as a synonym for dementia-related behavior in relation to possible BPSD. Aging people with ID may develop dementia at an earlier age and higher rate than the general population [27]. The prevalence of Alzheimer dementia (AD) is high among aging people with ID, especially those with Down syndrome [28–33]. However, the literature documents extremely heterogeneous AD rates depending on study designs and research populations [4].

A recent study on the signs of dementia among 230 people with ID aged from 34–80 years old, carried out using the British Present Psychiatric State–Learning Disabilities Assessment, found a prevalence of almost 42% with two or more signs and the overall frequency of symptoms, did not differ between age groups [5]. A Japanese survey of 493 aging people with ID confirmed increased prevalence rates of dementia relative to the normal population, of 3.5% among 55–64-year-olds and 13.9% among 65–74-year-olds [34]. A Swedish three-year follow-up study of 222 people with ID aged 60 years old or more found a prevalence of dementia of 15.7% or almost four times higher than the general population [35]. People with ID with Down syndrome are at the highest risk of developing AD, as demonstrated in numerous studies [6,31,32]. Approximately 15–40% of people with Down syndrome over the age of 35 present with a clinical picture of AD. Because the onset of dementia is so early, the average age of people with Down syndrome with AD is estimated to be 51.3 years old [32,33,36].

Pre-existing cognitive impairment makes the diagnosis of dementia in people with ID extremely difficult [37–39]. Initial levels of ID vary greatly from one individual to another and are often insufficiently documented [40]. The superimposed BPSD may be mistakenly attributed to ID, delaying dementia's diagnosis [41,42]. This cognitive deterioration is often

worsening the decline in executive functions [43]. Screening for dementia should follow a clear, detailed pattern to avoid overshadowing other conditions or problems and thus avoid under-reporting [44,45]. There is a negative correlation between levels of autonomy in the Activities of Daily Living and dementia in aging people with ID [46]. Among older adults with ID, BPSD are primarily detected following the development of irritability and aggressiveness. Among persons with Down syndrome, these specific symptoms more often lead to a psychiatric consultation than do psychological symptoms such as slowing down, apathy, loss of interest, drowsiness, irritation, fear, feeling sad, lack of appetite, and night-time restlessness [20,47,48]. However, direct support professionals are more likely to miss or misinterpret BPSD due to the characteristic overall clinical picture of ID [49,50]. As a result, direct support professionals sometimes fail to associate a new behavioral disorder with the underlying problem of BPSD [17]. The less intrusive or disruptive BPSD may be missed at first, and the real problem will only be recognized when the symptoms and signs become more pronounced [38,51].

Although direct support professionals do indeed identify behavioral disorders among aging people with ID, they have difficulty determining whether they are related to a known challenging behavior of ID or of BPSD in addition to ID [52]. Delivering adequate support and resources during these health transitions creates challenges for long-term care facilities and their direct support professionals [53]. Caring for aging people with ID with dementia is more complex and time-consuming than caring for people with ID without superimposed dementia [54]. Failure to recognize a clinical picture involving BPSD can lead to a late diagnosis of dementia or another dementia, which puts back optimal support. Enabling direct support professionals working with aging people with ID to improve their knowledge of dementia, detect BPSD, use best care practices, and optimize person-centered support will require better education in dementia care and the psychopathology of dementia. There have been few investigations of direct support professionals' experiences of providing daily support to aging people with ID presenting BPSD.

The study's primary aim was to investigate the perceived barriers to detecting, dealing with, and supporting people with ID presenting BPSD, as well as revealing the factors that might help direct support professionals optimize person-centered support. The second aim was to formulate recommendations in support of direct support professionals to help them provide best-practice care to aging people with ID presenting with BPSD. The following research questions guided our exploratory research: How do direct support professionals deal with aging people with ID presenting BPSD? What are their experiences and perceptions of the barriers to their work? Are there any associations between the sociodemographic and professional characteristics of direct support professionals and their ability to recognize BPSD described in the clinical vignettes? What might help them optimize their daily support?

## 2. Method

### 2.1. Design

This exploratory, multi-site, multi-methods research study used a cross-sectional clinical vignette study followed by qualitative research based on focus groups (FGs). Clinical vignettes were used to record how direct support professionals deal with aging people with ID presenting BPSD. FGs collected qualitative data on direct support professionals' lived experiences and perceived barriers to their support of people with ID presenting BPSD. It also investigated what might help them provide best-practice care in their daily support. We conducted a multi-method study within that project, guided by the Consolidated Criteria for Reporting Qualitative Studies (COREQ) [55] and the Strengthening the Reporting of Observational Studies in Epidemiology (STROBE) Statement [56].

### 2.2. Population and Setting

Three long-term care facilities for disabled persons in Switzerland's French-speaking region were invited to participate in the study. A long-term care facility can be described as

a skilled care facility providing a range of personal and health care services but focusing on medical, educational, and nursing care, and patient safety and educational support in the activities of daily living [57]. In close collaboration with those institutions' managements, their direct support professionals—men or women aged 18 years old or more and providing care to aging people with ID presenting with BPSD—were invited to participate in one of the FGs. Direct support professionals were defined as frontline professionals with a Bachelor's degree, college diploma, or professional education in social work or education qualifying them to take care of disabled persons. No exclusion criteria were applied for the recruitment of eligible direct support professionals' participants in the study.

## 2.3. Recruitment and Data Collection Strategy

The Human Research Ethics Committee of the Canton of Vaud (CER-VD) (2020-00141) approved the study for conduct between March and October 2020. Based on lists of eligible direct support professionals presented by participating long-term care facilities, the research team randomly selected 8 participants from each. Each eligible direct support professional received information about the study's objectives and nature, their expected participation, and their provisional appointment with the investigators. They were informed of the measures protecting their anonymity and rights. All the invited direct support professionals agreed to participate and signed the written consent form. Three groups of participants came to the appointment in their respective workplaces, responded independently to the clinical vignette study, and then sat in the FG together. FGs were conducted using pretested interview guides (Table 1), with groups of eight participants at each long-term care facility, and were audio-recorded. Participants' anonymity was ensured, and the study followed the Declaration of Helsinki's standards of good research practice. Although all three FGs were conducted in the direct support professionals respective long-term care facilities during working hours, they received no financial compensation for their participation.

**Table 1.** Semi-structured interview guide for the FGs of direct support professionals (*n* = 24).

| |
|---|
| What do you find problematic about supporting your residents on a day-to-day basis? Tell us about the most common problems you encounter when supporting people with ID presenting with new, unfamiliar behavior? |
| At what point do you estimate that a resident presents BPSD? What signs or symptoms do you look for? Which BPSD do you have the most difficulty identifying? |
| What do you do if one of your residents presents BPSD? What is your approach? Is your management different depending on the clinical form or manifestation of the behavior? |
| What specific knowledge and skills would help you accurately identify BPSD? What would you find helpful and supportive when dealing with occurrences of BPSD? |

## 2.4. Data Collection Instruments

### 2.4.1. Clinical Vignettes

To explore participants' lived experiences and management of BPSD among aging people with ID, each received three written clinical vignettes describing realistic clinical situations involving the onset of an unusual behavior superimposed on a usual ID challenging behavior. A clinical vignette is an abridged report about a patient, summarizing any relevant history, physical examination findings, data from investigations, and treatment information. Vignettes can be used for three main research purposes: to allow actions in context to be explored; to clarify people's judgements; and to provide less personal and therefore less threatening ways of exploring sensitive topics [58]. The clinical vignettes were based on a literature review [59], three days of field observation in the participating long-term care facilities, and a retrospective analysis of the aging people with ID's records. They were pretested on three volunteers to increase their relevance and readability. Based on the pretest group's feedback, minor modifications were made to the wording of the vi-

gnette's questions (Supplementary File S1). The following clinical vignettes were submitted to the participants:

- In clinical vignette 1, 67-year-old Louise suffers from a mental disability and psychosis and demonstrates increased cognitive disorders and functional decline requiring one-to-one care and support. She progressively refuses to take showers and medication.
- In clinical vignette 2, 60-year-old Eugenie is a care home resident with Down syndrome but no superimposed psychiatric disorders. Her behavior suddenly changed in just one day when she fell asleep repeatedly, alternating with periods of crying out and leading to agitated nights.
- In clinical vignette 3, 64-year-old Roger has Down syndrome but a high potential for conducting his activities of daily life autonomously. In just a few days he became more apathetic, resistant to care, refused to eat and drink, and demonstrated agitation at night.

All the responses to the clinical vignettes were based on the items in Table 3.

### 2.4.2. Focus Groups

FGs were defined as semi-structured discussions between eight direct support professionals with varied education trajectories. They explored their lived experiences, perceptions, and difficulties in supporting aging people with ID with BPSD, and they sought their perceptions of how to optimize support around occurrences of BPSD. The FG interview guide was pretested on two direct support professionals working at a similar long-term care facility. The FGs were facilitated by two researchers (K.E. and H.V.), and each FG began with a general question before moving to more specific ones. Table 1 presents the interview guide applied to the participants of the three FGs.

### 2.5. Data Analysis

The collected data were analyzed according to good clinical research practices. The clinical vignettes were analyzed based on the nine items described in Table 3. All the participants received the three clinical vignettes and their accompanying open questions, as mentioned in Supplementary File S1. The clinical vignette questionnaires were completed before the focus groups. The participants were asked to answer to the best of their knowledge and using their experiences in daily practice. Two authors (K.E. and H.V.) analyzed and coded the clinical vignettes based on a grid specially constructed for the study. If the written answer did not clearly mention that the behavior observed was related to the BPSD, then the evaluators considered this to be "unclear". Evaluations were made independently by two researchers (K.E. and H.V.), with disagreements resolved by discussion.

Descriptive, comparative, and association statistics were computed to explore the distributions, differences, and relationships around recognizing BPSD in the clinical vignettes. Data were imported into Statistical Package for the Social Sciences (SPSS) software, version 27.0 (IBM-SPSS Inc., Chicago, IL, USA), for analysis. The significance level for all the analyses was set at $p < 0.05$.

Audio recordings from each FG (FG 1, FG 2, and FG 3) were transcribed verbatim and uploaded into NVivo 12© (QSR international software, Daresbury, Cheshire, UK). Data were condensed using the interview guide's themes. Phenomenological content analysis focused on participants' lived experiences in supporting older adult people with ID presenting with BPSD. Analyses emphasized their cumulative viewpoints in response to the research questions. Data analyses followed a three-step process. First, to promote immersion in the data, researchers (K.E. and H.V.) read the transcripts from the three FGs over several days. Second, responses were coded using NVivo 12® in semantic nodes based on the interview guide.

Third, concept mapping was used to group the nodes into conceptual themes. Several strategies were employed to increase the rigor of the data analyses and the reliability of the findings [60]. Specifically, the research team retraced the coding and concept-mapping

procedures to ensure that the conclusions followed a sensible path emanating directly and accurately from the direct support professionals' members' narratives.

## 3. Results

### 3.1. Participation and Sociodemographic Characteristics

All 24 of the direct support professionals recruited agreed to participate in the study and gave their oral and written consent. Two-thirds of participants were women, their average age was 35 years old, and they had an average of almost six years of professional experience (Table 2). Participants' educational trajectories were heterogenous: about 40% were non-university-educated direct support professionals with a traditional secondary school education; the remaining 60% had undergone university social or educational training (Bachelor's or Master's degree) (Table 2). This variability gave rise to rich exchanges of experiences in supporting aging people with ID and dementia-related behavioral disorders.

**Table 2.** Participants' sociodemographic and professional characteristics (*n* = 24).

| Participants | Data (*n* = 24) |
|---|:---:|
| Sex | |
| Male/Female | 8/16 |
| Age | |
| Mean (SD) | 35.2 (10.8) |
| Median (IQR-75) | 32 (46.7) |
| Min–Max | 21–55 |
| Educational level | |
| Non-university-educated professionals (%) [a] | 10 (41.7) |
| University-educated professionals (%) [b] | 14 (58.3) |
| Years of professional experience | |
| Mean (SD) | 5.2 (4.6) |
| Median (IQR-75) | 3.0 (9.3) |
| Min–Max. | 1–16 |

Note: [a] non-university-educated professionals (secondary school-educated professionals); [b] university-educated professionals (psychologists, specialized educators).

### 3.2. Clinical Vignettes

3.2.1. Descriptive Results

All the participants responded to three clinical vignettes (Table 3), spending from 25–40 min to do so (mean = 31; SD = 6). Less than half of the participants identified the onset of BPSD in the first clinical vignette. Three-quarters of participants' responses were evaluated as unclear by K.E. and H.V. about whether the situation was due to behavior typical of an ID or BPSD and wished to check their observations with other team members. None of the participants mentioned using specific observation tools to document the occurrence of BPSD. Despite the difficulties in identifying occurrences of BPSD, most participants were well trained in how to proceed and organize a follow-up linked to these behavior changes by involving nurses and the people with ID's family or a specialist physician. One-fifth of the participants were able to identify the occurrence of BPSD in the second vignette. However, almost half of participating direct support professionals responded unclearly (K.E. and H.V. evaluation), not knowing whether the behavior was due to ID or BPSD; three participants did not respond to the vignette and gave no reason. In the third vignette, almost two-thirds of the participants scored better, recognizing the occurrence of BPSD described. However, about one-fourth were still unclear about their assessment of the clinical event. Although responses to the third vignette were considerably better than to the first two, none of the participants mentioned documenting occurrences of BPSD using a specific measurement (type of behavior, time, frequency, severity, and effects of the residents' environment). Table 3 presents a detailed overview of the participants' responses to the clinical vignettes.

**Table 3.** Distribution of participants' responses to the clinical vignettes, evaluated by two researchers (K.E. and H.V.) (*n* = 24).

| Items | Vignette 1: *n* (%) | | | | Vignette 2: *n* (%) | | | | Vignette 3: *n* (%) | | | |
|---|---|---|---|---|---|---|---|---|---|---|---|---|
| | Yes | No | Unclear | No Answer | Yes | No | Unclear | No Answer | Yes | No | Unclear | No Answer |
| Was the situation identified as an unusual change in behavior related to dementia? (*n* = 23) | 2 (8.3) | 5 (20.8) | 16 (66.7) | 1 (4.2) | 6 (25.0) | 2 (8.3) | 15 (62.5) | 1 (4.2) | 9 (37.5) | 0 (0.0) | 13 (54.2) | 2 (8.3) |
| Did the participant understand the situation? (*n* = 23) | 4 (16.7) | 1 (4.2) | 18 (75.0) | 1 (4.2) | 10 (41.7) | 1 (4.2) | 12 (50.0) | 1 (4.2) | 15 (62.5) | 0 (0.0) | 7 (29.2) | 2 (8.3) |
| Was the approach proposed well thought out/in accordance with good practice? (*n* = 23) | 17 (70.8) | 0 (0.0) | 6 (25.0) | 1 (4.2) | 14 (58.3) | 2 (8.3) | 7 (29.2) | 1 (4.2) | 14 (58.3) | 0 (0.0) | 7 (29.2) | 3 (12.5) |
| Was the approach structured and pragmatically implemented? (*n* = 23) | 16 (66.7) | 0 (0.0) | 7 (29.2) | 1 (4.2) | 14 (58.3) | 2 (8.3) | 7 (29.2) | 1 (4.2) | 14 (58.3) | 0 (0.0) | 7 (29.2) | 3 (12.5) |
| Did the participant propose involving other colleagues? (*n* = 23) | 18 (75.0) | 0 (0.0) | 4 (16.7) | 1 (4.2) | 13 (54.2) | 1 (4.2) | 7 (29.2) | 3 (12.5) | 14 (58.3) | 0 (0.0) | 6 (25.0) | 4 (16.7) |
| Was the proposal to involve other colleagues relevant or justified? (*n* = 23) | 16 (66.7) | 1 (4.2) | 5 (20.8) | 1 (4.2.) | 11 (45.8) | 1 (4.2) | 9 (37.5) | 3 (12.5) | 14 (58.3) | 0 (0.0) | 6 (25.0) | 4 (16.7) |
| The participant mentioned the difficulties of dealing with this situation (*n* = 23) | 16 (66.7) | 1 (4.2) | 5 (20.8) | 2 (8.3) | 11 (45.8) | 1 (4.2) | 9 (37.5) | 3 (12.5) | 14 (58.3) | 0 (0.0) | 6 (25.0) | 4 (16.7) |
| Did the participant mention any barriers to dealing with this situation? (*n* = 23) | 15 (62.5) | 0 (0.0) | 7 (29.2) | 2 (8.3) | 11 (45.8) | 1 (4.2) | 9 (37.5) | 3 (12.5) | 14 (58.3) | 0 (0.0) | 6 (25.0) | 4 (16.7) |
| Did the participant propose any solutions? (*n* = 23) | 14 (58.3) | 0 (0.0) | 8 (33.3) | 2 (8.3) | 11 (45.8) | 1 (4.2) | 9 (37.5) | 3 (12.5) | 14 (58.3) | 0 (0.0) | 6 (25.0) | 4 (16.7) |

Thirteen participants (54%) did not recognize a single occurrence of BPSD in the three clinical vignettes, five (21%) recognized one occurrence, and six (25%) recognized two occurrences of BPSD. None recognized all three occurrences of BPSD described. Except for age, there were no significant differences between participants' sociodemographic and professional characteristics and their ability to recognize occurrences of BPSD in the clinical vignettes—older direct support professionals detected more occurrences of BPSD than younger ones (*p* = 0.022) in the second clinical vignette (Table 4).

**Table 4.** Comparison of sociodemographic and professional characteristics and recognition of BPSD (*n* = 24).

| Sociodemographic and Professional Characteristics | Vignette 1 Recognized BPSD Yes/No | *p*-Value | Vignette 2 Recognized BPSD Yes/No | *p*-Value | Vignette 3 Recognized BPSD Yes/No | *p*-Value |
|---|---|---|---|---|---|---|
| Sex | | | | | | |
| M | 1/7 | | 1/7 | | 2/6 | |
| F | 1/15 | 0.565 [a] | 5/11 | 0.319 [a] | 7/9 | 0.332 [a] |
| Age | | 0.652 [b] | | 0.022 [b,*] | | 0.558 [b] |
| Education level | | | | | | |
| Non-university-educated | 1/9 | | 4/6 | | 3/7 | |
| University-educated | 1/13 | 0.670 [a] | 2/12 | 0.170 | 6/8 | 0.418 [a] |
| Years of professional experience | | | | | | |
| Median (IQR-75) | | 0.877 [b] | | 0.343 [b] | | 0.682 [b] |

Note. [a] Fisher's exact test; [b] Mann–Whitney U test; * Significant < 0.05.

3.2.2. Associations between Sociodemographic and Professional Characteristics and Recognition of BPSD

Except for age having a moderately significant positive association in the second vignette (T = 0.393; *p* = 0.05; older participants were more able to recognize BPSD), no other

significant associations were found. Table 5 present the associations between sociodemographic and professional characteristics and recognizing BPSD in the clinical vignettes.

**Table 5.** Associations between sociodemographic and professional characteristics and vignettes.

| Sociodemographic and Professional Characteristics | Vignette 1 | Vignette 2 | Vignette 3 |
|---|---|---|---|
| Sex | 0.107 [a] | 0.204 [a] | 0.183 [a] |
| Age | −0.092 [b] | 0.393 *,[b] | 0.105 [b] |
| Educational level | 0.051 [a] | 0.293 [a] | 0.131 [a] |
| Years of experience | −0.029 [b] | 0.179 [b] | 0.077 [b] |

Note. * significant association $p < 0.05$; [a] Cramér's V test; [b] Kendall's Tau test.

### 3.3. Focus Group Findings

Twenty-four direct support professionals participated in one of the three focus groups in the long-term care facilities. A total of 3 h and 23 min of discussion were recorded (mean = 67 min; SD = 13; median = 67; min. = 64; max. = 72). Participants exchanged their thoughts and experiences of practicing with aging people with ID exhibiting challenging behaviors and superimposed BPSD. The themes that emerged included barriers to caring for aging people with ID, difficulties identifying BPSD in aging people with ID, and the need to increase staff knowledge and skills to optimize person-centered care for aging people with ID presenting with BPSD. Participants also expressed their concerns for aging people with ID and mentioned the need for interdisciplinary teams with the mix of skills needed to provide competent clinical knowledge and better-adapted care. Each theme was generated from patterns in the discussions' core content. Table 6 present the summary of the results of the focus groups.

**Table 6.** Summary of FGs and the core content of the themes.

| Themes | Content of Focus Groups Discussions |
|---|---|
| Difficulties caring for aging people with ID presenting with BPSD | - Direct support professionals are distressed by occurrences of BPSD and experience difficulties managing them<br>- Transfer of clinical practice skills is based on the empirical experiences of more experienced direct support professionals<br>- The basic educational methods used with aging people with ID presenting with BPSD are frequently ineffective |
| A close care relationship provides time for reflection and understanding behavioral changes | - Direct support professionals stated that being able to identify the subtle changes towards unusual behaviors was critical<br>- Close care relationships allow direct support professionals to detect subtle changes in autonomy and cognitive abilities |
| Comprehension and detection of BPSD in daily practice | - Direct support professionals are unclear how to discriminate between the challenging behaviors of ID and BPSD<br>- The onset of BPSD affects everyday life and makes it problematic for direct support professionals to understand what is happening |
| Difficulties in implementing management strategies for dealing with BPSD | - Difficulties identifying BPSD and attributing it to the challenging behaviors of ID<br>- Pharmacological and psychosocial approaches to BPSD are not applied<br>- The absence of a diagnosis of dementia complexifies the detection of BPSD and delays the implementation of best practices to manage them<br>- There is a lack of awareness about tools and strategies to assess and implement psychosocial approaches to managing occurrences of BPSD |
| Need to be able to accurately identify BPSD so as to deal with aging people with ID presenting with them | - Need to increase knowledge about psychiatric, psychogeriatric, and geriatric diseases and syndromes<br>- Need for more training and education based on clinical case studies and evidence-based practice |

### 3.3.1. Difficulties Caring for Aging People with ID Exhibiting Challenging Behaviors and Superimposed BPSD

Participants noted that deteriorating health was highly prevalent among aging people with ID, and they requested updates on their knowledge and competencies to manage the atypical manifestations of geriatric syndromes, including pain detection, palliative care, falls, loss of autonomy, and cognitive impairment superimposed on challenging behaviors related to ID. Participants reported BPSD as "unusual behavior" and designated them as significant problems in daily care. They mentioned being distressed by new occurrences of BPSD and experiencing difficulties in managing these health issues, such as the onset of agitation, opposition to care, or disruptive vocalization. Participants explained that their clinical practice was mainly based on transfers of the empirical experiences of more-experienced direct support professionals to less-experienced direct support professionals. Participants highlighted that current care approaches were based more on intuition than scientific knowledge or trained skills built on best practices. Most direct support professionals applied basic educational methods to their daily practice with aging people with ID presenting with BPSD, but they noted that these experiences were frequently ineffective. The following extract is typical of participants' opinions on geriatric health issues and the daily care of aging people with ID exhibiting BPSD:

*"I think that going beyond maintenance [of their skills], as you said, we're an educational team, and we're part of a network of educational institutions. It's not just about maintenance; there are also residents with real needs because of what they are living through. Sometimes they're people who've known other residents, people who were around them at various times in their lives, so they have a representation of those people as alive—I mean other than them being unable to walk anymore, not being able to eat, not being able to speak. In fact, that emotional, psycho-affective dimension is very frustrating because their needs are massive. It's nonetheless important, in my view, just as important as supporting someone who needs help to eat and sleep, or with personal care, for example, with incontinence and all that. I don't think that it's less important than somebody who has primary health needs."*

(FG1)

*"Well, me, I'd say it's like there were two different worlds. That's to say, there's . . . us; we're educators, with our vision of things and support as educators. And everything we're going to do is already set within that vision, if you like. But we're tinkering about, and there's a lack of—I don't know if it's training or the transmission of information or simply just a daily mix up of things that mean that in these situations we've already got things available, in advance, somehow. And that's true when we get to the medical side of things, or they call on us and give us something to put in place, its . . . They come and tell us how to do things. But then that vision doesn't fit with ours. So that creates a conflict, and there's one resident who sometimes gets forgotten in all that. So . . . because we just tinker away, the others will come and tell us, 'We've got our vision of things.'"*

(FG3).

### 3.3.2. Close Care Relationships Provide Time for Reflection and Understanding Behavioral Changes

The participants expressed how critical it was to identify subtle changes in usual behaviors, indications of the first symptoms, and signs of cognitive decline and BPSD. A close care relationship, facilitated by adequate staffing and care management tools, guides how daily care and support are provided, according to changing needs and new circumstances, and allows direct support professionals to detect subtle changes in autonomy and cognitive abilities. Participants described the importance of long-term care relationships and how the behavioral changes that might signal dementia usually required direct support professionals to strive to understand people with IDs' changing behaviors. The following transcript illustrates those concerns:

*"Behaviors are very diverse, actually. And maybe that's why it's so difficult, because it's never the same. And we never have time to find out. Well, we don't always have the time to follow up on the right leads. By the time we identify them, the behaviors have changed again. So we're constantly searching for the right behavior. It evolves very fast."*

(FG 1)

*"I can't say how many years ago it was. His behavior just changed completely, and we didn't understand a thing. He'd been the nicest man in the world, and then he just changed completely."*

(FG2)

### 3.3.3. Understanding and Detecting BPSD in Daily Practice

Participants reported numerous lived experiences of unusual behavior or BPSD in the daily care of aging people with ID. Most of the direct support professionals expressed how difficult it was to attribute the unusual behavior to challenging behavior linked to ID or to BPSD. They also mentioned that occurrences of BPSD, especially agitation, aggression, vocalization, and opposition to care or medication, were complex and often tricky to deal with. Participants also spoke about transitions from care relationships with no symptoms and signs of dementia to care relationships where these had been detected and how this affected care goals and the promotion of autonomy. Onset BPSD in aging people with ID affect their everyday life and activities, especially during meals: forgetfulness makes it harder for aging people with ID to orientate themselves in time and finish meals. Direct support professionals need to be able to understand what is happening, that things need to be adapted, and how to adapt them. They expressed their compassion and concern about changes in the behavior and everyday habits of their patients, as illustrated in the following transcript:

*"For me, it's the resistance. Resistance to . . . so, it can be resistance to care, resistance to eating. It's actually anything that we know will have a negative impact on the person's health. Even resistance to going to the toilet when the [incontinence pants] are full of urine, for example. There you go. After a while, you know that you're going to have to get them there, for the person's own good. I think that's it."*

(FG 2)

*" . . . I think it's the residents' difficulties communicating—well, that's what's complicated because, you know, you are always assuming something, because they can't express what they are feeling, you know, emotionally, physically, and so you are always hypothesizing, and you ask yourself what you can do to support them, who you can contact to guide you, and you don't necessarily have a list of people to contact for each problem. So, there you go."*

(FG 3)

### 3.3.4. Difficulties in Implementing Management Strategies to Deal with BPSD

Participants expressed difficulties identifying unusual behavior and attributing it to the challenging behaviors of ID. Most participants called upon nursing and medical staff to help them deal with BPSD. Some participants expressed feelings of "failing" in the management of unusual behavior for different reasons. Participants mentioned that pharmacological approaches were not always applied in cases involving BPSD. Direct support professionals mentioned several reasons why they did not use pharmacological approaches, such as a lack of knowledge about psychotropic drugs and their effects on behavior, and they interpreted them as surrogates for educational approaches. The absence of a probable diagnosis of dementia among aging people with ID caused participants to have doubts about attributing behavior to dementia and led to delays in introducing best practices for managing BPSD. The participants also mentioned their lack of knowledge about the clinical criteria for diagnosing dementia and not being aware of tools or strategies

for assessing and implementing psychosocial approaches for managing occurrences of any BPSD. The following transcript reveals some of their difficulties:

*"In fact, it depends on which one. No, it depends on the person; it depends on the behavioral disorder. Well, instead, I'll start out observing, reassuring, well, trying to support the person at the time and observing what's going on in the group. The somatic stuff comes later, actually, if we see that it happens several times. Well, for the somatic stuff, you do ask yourself the question at the time, but you can't resolve it right away; it's not immediate."*

(FG 3)

*" . . . think it's dangerous, that feeling of failure when it's [time for anticipatory drugs], because often too many medications aren't a good thing. But if medicine can calm someone down, it depends on the drug. I tell myself, 'If that anticipatory drug is for knocking somebody out, then I don't see the point.' That's it. I guess that sometimes a medicine can bring relief to the person concerned, something that we can't do, you know. So . . . yes, I understand the question."*

(FG 1)

### 3.3.5. The Need for Accuracy When Dealing with Aging People with ID Presenting with BPSD

Participants believed that it was essential that they should have more knowledge about the psychiatric, psychogeriatric, and geriatric diseases and syndromes prevalent among aging people with ID. Although most of the participants stated that their patients regularly presented with unusual behavioral disorders or the BPSD, it was sometimes difficult to recognize them, especially because they lacked the right knowledge. Participants also mentioned the heterogenicity of the clinical pictures of BPSD among aging people with ID as a critical issue. They mentioned their need for more knowledge by evoking examples of the different behaviors exhibited by persons with Down syndrome who also had dementia and by aging people with ID with dementia. Finally, they expressed their need for more training and education based on case studies and best practices. The following transcript illustrates the participants' concerns:

*"Actually, we don't have enough hindsight. It's not been long. In fact, it's new: aging linked to disability is a new challenge. And, yes, there are studies on it; yes, that's starting; we're starting to ask ourselves questions about it. We haven't got the necessary experience, and we don't have any, we get some information on the fly, and we don't really have any tools. What's missing for me, actually, is tools—for [supporting] these people, to study them—and the lack of specialists too. For example, support during bereavement—I've looked into this question quite a bit—anyway, yes, they're people who experience mourning like anyone else. Maybe when it comes to tools, we could support them through this differently than for the average person. Anyway, it's things like that. So, yes, we need more research and tools."*

(FG 1)

*"It's true that depending on the population we're hosting, we'll orient our hiring more specifically towards staff trained to guide groups. So, that's a first thing. Then again, that's not necessarily any guarantee of success or support, either. So, support requires a group of people, with several actors, and not just one team, I think. Thus, support from the health unit is primordial in this sort of thing. I think that the communication dynamics within the team is very important too. I think that it is important to be able to discuss things with those people. And I think we are getting into situations that are a bit new for our institutions. And then I'm not sure that we've looked into these questions deeply enough. The proof of that is that you're here today. So, I haven't got a ready-made response to give you. We'll only get more effective by sharing our competencies and experiences—with humility. Well, we'll be less awful; because there are times that there*

*are architectural, structural constraints that limit the support we can give. It's not always the educational team ['s limits]."*

(FG 2)

## 4. Discussion

The present study aimed to collect data from direct support professionals about barriers to their daily practice and support of aging people with ID presenting with BPSD. The project focused on recognizing occurrences of the BPSD superimposed on the challenging behaviors of ID as described in clinical vignettes and collecting the lived experiences and perceptions of taking care of aging people with ID with unusual behavior. The managements of three long-term care facilities hosting aging people with ID were enthusiastic about participating in this investigation. They indicated that the topic of dementia and BPSD was a growing concern with regards to their residents. This was confirmed by the direct support professionals who participated and stated that the topic was indeed relevant and that the difficulties supporting aging people with ID were an emerging issue. Moreover, based on our results, the multi-method design selected worked well to reveal the barriers direct support professionals perceived in their support for aging people with ID presenting with BPSD. The research population, composed of frontline professionals, corroborated most of the previous investigations in long-term care facilities hosting aging people with ID [61].

Our findings demonstrated that most direct support professionals had difficulties to recognize BPSD superimposed on the already challenging behavior of ID. This result was worrying, bearing in mind that recent studies have highlighted—even after excluding persons with Down syndrome [5,62–64]—high prevalence rates (>40%) of two or more signs of AD among people with ID from 34–80 years old, and the frequency of symptoms did not differ between different age groups within that range or between numerous rare syndromes with comorbidities [5,34]. Besides people with Down syndrome presenting with an increased risk of dementia, people with other genetic syndromes, such as Sanfilippo, Williams, and Fragile X syndromes, have also been identified as having higher epidemiological rates [41,65,66]. This underlined that diagnosing dementia in people with ID is more complex than in the general population due to varying levels of pre-existing intellectual impairment and the increased presence of health conditions or behaviors that often mimic the symptoms of dementia. This was in line with the few previous studies conducted on this topic [61].

Direct support professionals stated that the detection of dementia among aging people with ID was complex but was an essential part of preventing or managing BPSD. They suggested that increasing their knowledge and being informed of probable diagnoses of dementia would help them optimize dementia detection and management of BPSD. These results are consistent with Cleary and Doody's, Iacono's et al. and Dekkers et al. results, which pointed out that knowledge of a probable diagnosis of dementia in aging people with ID facilitated the detection of behavioral changes and the optimization of support strategies [19,67,68]. This highlights the importance of improving the early detection of dementia in long-term care facilities hosting aging people with ID. Moran et al. suggested using a broader diagnostic workup for people with ID, including detailed histories presented by someone well-acquainted with the individual, such as a family member and a professional caregiver [69]. Most of the participants were unaware of the prevalence of dementia among aging people with ID presenting with BPSD, something making any understanding of the presence of unusual behavior far more complex. This is surprising considering that the consensus criteria for dementia have been developed for people with ID [70]. However, given the pre-existing cognitive impairments among people with ID, it makes the dementia situation extremely difficult for direct support professionals to distinguish between challenging behaviors related to ID or superimposed BPSD [2,22,71].

Furthermore, some tools have recently been developed, such as the Cambridge Examination for Mental Disorders of Older People with Down Syndrome and Others with

Intellectual Disabilities (CAMDEX-DS), to explore people with ID with Down syndrome and changes in the dementia-related domains of cognition and function, the test battery for the diagnosis of dementia in individuals with intellectual disability de Burt & Aylward and the Dementia Screening Questionnaire for Individuals with Intellectual Disabilities of Deb et al. [72–74]. A recent systematic review by Zeilinger et al. (2022), including 42 eligible publications, summarized the existing 'Informant-based assessment instruments for dementia in people with intellectual disability'. About 18 informant-based assessment instruments for screening were found. These authors recommended the Behavioral and Psychological Symptoms of Dementia in Down Syndrome Scale (BPSD-DS), the Cognitive Scale for Down Syndrome (CS-DS), and the Dementia Screening Questionnaire for Individuals with Intellectual Disabilities (DSQIID). For a more thorough dementia assessment, we recommend the Cambridge Examination for Mental Disorders of Older People with Down's Syndrome and Others with Intellectual Disabilities (CAMDEX-DS) [25].

Another interesting avenue could be specialized memory clinics for people with IDs with possible dementia, using informant-based tools and objective-based tools for measuring possible dementia and evaluating the dementia process [5,75,76]. One interesting result from the present study was that professional experience did not increase the ability to recognize occurrences of the BPSD among aging people with ID. Our findings about the connection between experience and knowledge is not in line with the findings of Holst et al., who found that more experienced educational staff were better at recognizing symptoms of dementia [17]. Our study suggests that given the appropriate training, newer employees can attain the same level of knowledge as their colleagues with many years of experience.

Direct support professionals endorsed the competencies of their long-term care facilities' healthcare staff in supporting them with occurrences of BPSD. In the past, care for people with IDs focused mainly on the socio-educational efforts to promote independence and choice [77,78], activities mainly carried out by direct support professionals [79]. Participating frontline educational staff showed a high level of commitment and motivation for their work. However, they also expressed the desire for more knowledge and training, as this was almost a prerequisite for them to be able to continue supporting people with ID, as progressing dementia increased the need for care. Indeed, the International Summit on Intellectual Disability and Dementia concluded that new skills and knowledge on aging people with ID and cognitive disorders were essential for direct support professionals [80]. In terms of knowledge about dementia, our findings indicated that most respondents had gained their knowledge through lived experiences in the workplace rather than continuing education or training. Direct support professionals working in long-term care facilities for people with ID generally have a social or educational career trajectory, meaning they may have no specific knowledge about dementia other than from their experiences at work. Previous studies demonstrated that direct support professionals generally expressed uncertainty about dementia's trajectories, boundaries between age-related memory decline, cognitive impairment, and dementia [17]. Janicki et al. suggested that staff education and training were crucial components of the quality of care for people with ID and dementia [81,82].

This will require continuing education about dementia and dementia-related behavior disorders for staff that is planned and adapted to meet the changing needs of aging people with ID, as Wilkinson et al. and Holst et al. [17,83] have suggested. Olsson et al. and Truong et al. found that after receiving an educational intervention, staff's working knowledge and attitudes towards people with IDs had significantly improved and they were better prepared for differentiating between various mental health problems [61,84]. Our findings also indicate that longitudinal clinical assessments and investigation of dementia should be planned among people with ID involving direct support professionals, as stated by the International Summit on Intellectual Disability and Dementia [80]. It is well known that diagnosing dementia in people with ID is complex due to their intellectual and psychosocial deficits and, very often, the atypical presentation [71]. Additionally, our results suggest that it is critical to build interdisciplinary teams and specialized long-term care units for people with ID and dementia to optimize person-centered evidence-based care.

From our findings, outlined above, it seems reasonable to consider that early onset dementia as an important risk among aging people with ID. There is a need for additional representational studies on epidemiology and more clinicopathological correlation studies to provide evidence on which to base education and recommendations for caregivers and direct support professionals. There will also be a need for projects to develop and evaluate interventions to manage or prevent BPSD in people with ID. Although prevalence studies have identified that certain medical conditions and risk factors for dementia and its associated behaviors occur more frequently in people with ID, there are currently no national clinical guidelines on the best practices for supporting residents in long-term care facilities. Indeed, there have recently been calls to include people with ID as a specific population requiring distinct consideration in standard clinical guidelines [85].

Investment in educational programs on dementia and clinical practice should primarily aim to improve the skills and abilities of direct support professionals and thus improve the quality of care for people with ID at risk of dementia and occurrences of BPSD.

### 4.1. Study Strengths and Limitations

To the best of our knowledge, this is a first investigation exploring the barriers direct support professionals face in recognizing occurrences of BPSD among people with ID in long-term care facilities in the French-speaking region of Switzerland. Although this exploratory, mixed-methods study was rigorously conducted, the vignette study's small sample size suggests that results should be interpreted with caution. Quantitative analysis was limited by the small sample, which only enabled basic, dichotomic cross-tabulation statistical tests. The study's limited length and resources made it impossible to conduct field investigations to directly observe occurrences of BPSD among people with ID and how direct support professionals provided care. Additionally, the study was unable to explore any existing historical data enabling the detection of dementia (Dementia Screening Questionnaire for Individuals with Intellectual Disabilities) or question family and direct care staff caring for people with ID and dementia.

It would have been very interesting and pertinent to have involved people with ID in the study, letting them express their thoughts and wishes autonomously, and this may have illuminated the process of recognizing the first signs of dementia. However, since this would imply significant methodological and ethical difficulties, direct support professionals can be considered their representatives in this study.

The direct support professionals focus groups may give a somewhat fatalistic picture of the daily care for people with ID presenting with BPSD, but their lack of knowledge about dementia in general and AD in particular, and other common health problems in aging people with ID, should be considered when reading the results. However, our analysis was based on extensive narratives and data saturation; thus, interviewees' overall lived experiences can count as relevant internal validity for the study. Dementia among people with ID was not originally the central question in our interview guides, but it became the most apparent concern that came to light. The data collected reached saturation level and, consequently, our findings may be transferable to direct support professionals working in other Swiss long-term care facilities hosting aging people with ID. Based on federal laws, most long-term care facilities in Switzerland are organized and provide care similarly, based on the common core values of autonomy, inclusion, and participation. However, the transferability of our findings to other countries should be considered carefully.

### 4.2. Recommendations

With the current demographic transition and increasing life expectancy for aging people with ID and possible dementia, there will be an undeniable need to better prepare specialized institutions and care professionals by strengthening their knowledge and skills to support optimal aging despite very complex health trajectories. The present study indicated that our participating direct support professionals need and want to acquire that knowledge. Direct support professionals have unique knowledge about practicing care in

the specific contexts of their workplaces and with particular residents, but their knowledge needs to be updated with the latest evidence. Proper support and education would complement direct support professionals' essential experiential knowledge of dementia and introduce best practices for dealing with occurrences of BPSD. This should be coherent with their professional values, which already rely strongly on their knowledge about the challenging and unusual behaviors related to ID. Therefore, there is an urgent need for direct support professionals to receive information and training about the early signs of dementia and best practices for dealing with occurrences of BPSD among people with ID and telling them apart from ID behaviors. In highlighting direct support professionals' perspectives and practices, it is crucial to recognize training's role in their work and their preferences for that training. Based on our findings, we recommend the following:

- Systematic training programs for all staff providing support to people with ID who are approaching middle age. Lessons learnt should be applied before anyone develops dementia.
- Although approximately 15–40% of people with Down syndrome over the age of 35 present with a clinical picture of AD, because the onset of dementia is so early, the average age of people with Down syndrome and AD is estimated to be 51.3 years old. Guidelines on diagnoses, care pathways, and baseline assessments should be used with people with Down syndrome from the age of 35.
- Training programs for direct support professionals should include: a thorough explanation of what dementia in general and dementia in particular is; how differential diagnoses of clinical manifestations of the usual challenging behaviors linked to ID; descriptions of the objective experiences and subjective realities of people with dementia; how BPSD affect communication; how to develop suitable care environments; maintaining skills and developing appropriate activities; and information on medication, mobility issues, recognizing pain and managing it, supporting people with ID in eating (particularly issues concerning swallowing), and palliative care or end-of-life support.
- Based on current knowledge and promising psychometric properties, we recommend the use of appropriate, validated behavioral scales or questionnaires to aid direct support professionals and nursing staff in documenting unusual dementia-related behavior.

## 5. Conclusions

This study explored the barriers direct support professionals perceived to their being able to recognize the signs of suspected dementia and deal with BPSD among people with ID. Our findings highlighted the importance of dementia symptomatology and managing BPSD when supporting aging people with ID. Furthermore, our results underlined the importance of care relationships built up consistently over time. Only when relevant knowledge transfer occurs through continuous education/training and proactive dementia detection is implemented will it be possible for staff to identify dementia, react to its symptoms safely based on knowledge and evidence, initiate examinations, and design well-adapted care. In addition, close collaboration with dementia care specialists—and their skilled guidance and support—may contribute to better working environments for direct support professionals; together with continuous education and the initiation of intervention research concerning the best accommodation form for people with ID and the direct support professionals.

Our findings are key for policymakers aiming to improve the support provided to this particularly vulnerable group. There will have to be a greater focus on teaching direct support professionals about dementia among aging people with ID, since this critical area is inadequately covered today. In addition to staff experiences, further epidemiological and qualitative research might include perspectives on the early signs of dementia from people with ID themselves as well as their family members' experiences. This would also help direct support professionals and family members acquire a more profound knowledge of detecting and distinguishing signs of dementia over time and explore the living standards and technological environment.

**Supplementary Materials:** The following supporting information can be downloaded at: https://www.mdpi.com/article/10.3390/disabilities2040047/s1, File S1: Your approach to dealing with the situations described in the following clinical vignettes is of great interest to us.

**Author Contributions:** K.E. and H.V. designed the study in collaboration with A.v.G., V.G., D.G., T.B., and D.M. K.E. and H.V. ran the focus groups. K.E., T.B., D.M. and H.V. did the qualitative analysis. K.E., V.G., D.G. and H.V. wrote the manuscript's first draft. All authors commented on the results of the analysis and improved the draft manuscript. All authors approved the manuscript's final version. K.E. and A.v.G. were responsible for acquiring funding. All authors have read and agreed to the published version of the manuscript.

**Funding:** The study was supported with a grant from Lausanne University Hospital's Service of Old Age Psychiatry.

**Institutional Review Board Statement:** The study was conducted in accordance with the Declaration of Helsinki, and approved by the Human Research Ethics Committee of the Canton of Vaud (CER-VD) (2020-00141) for conduct between March and October 2020.

**Informed Consent Statement:** Informed consent was obtained from all participants involved in the study.

**Data Availability Statement:** The dataset used and analyzed during the current study is available from the corresponding author on reasonable request.

**Acknowledgments:** We would like to thank the long-term care facilities and all the direct support professionals who participated in this study and the long-term care facilities for their excellent collaboration and support. We also acknowledge the financial support of Lausanne University Hospital's Service of Old Age Psychiatry.

**Conflicts of Interest:** The authors declare no conflict of interest. The sponsor had no role in study design, data collection, analysis or interpretation, writing the manuscript, or the decision to publish the results.

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
