# Peer review of "Barriers Facing Direct Support Professionals When Supporting Older Adults Presenting with Intellectual Disabilities and Unusual Dementia-Related Behavior: A Multi-Site, Multi-Methods Study"

_disabilities, doi:10.3390/disabilities2040047_

Round 1
Reviewer 1 Report
This is an important study about how caregivers perceive BPSD in people with ID. Study outcomes are relevant and timely. However, the paper needs revision in a number of major aspects:
- Introduction and discussion should be shortened substantially. The key message is lost due to all side details.
- Method could be elaborated upon
- Referencing should be improved. A number of recent studies (especially in the field of down syndrome) is lacking.
Abstract:
- Abbreviation PWID is not common to use. Although it may save words, people in the disability community generally don't appreciate this. Please use either people with ID or people with intellectual disabilities in full.
Introduction:
In the general, the introduction is way too long. It reads like a text-book with a lot of general information and background that is not the essence of this study, among others about definitions of ID, prevalence of AD etc. Although many references are used, a number of recent key publications is missing.
- What is meant with frontline educational staff? This term, as well as its abbreviation FES is very uncommon. How is frontline defined? Don't the same issues present for other staff? Please use 'caregivers' or 'direct support professionals'.
- Line 44: 'level of impairment' should be 'level of disability'
- Line 46: please provide the average life expectancy for people with Down syndrome
- Line 52: the authors state "The degree 50 of impairment—mild, moderate, severe, or profound/complete—is determined by 51 adaptive functioning rather than the intelligence quotient" It might be somewhat confusing, because although adaptive functioning is now regarded more important in defining the level of ID, cognitive (intellectual) functioning needs to be considered as well. IQ is thus no longer leading, but should not be ignored. Please rephrase. Moreover, the authors should realise that in daily practice, IQ is still often being used.
- Page 2: the introduction is rather long and text-book like. Why describe all definitions of intellectual disability? This is not the subject of the article.
- Page 2: the authors should use paragraphs. The text is not well-structured and too long.
- Line 77/78: Behavioral changes and functional decline compared to a person’s 77 previous functional abilities are the first signs and symptoms of superimposed dementia 78 [23]. The authors refer to ref. 23 here, which is an interview study with staff. It would be appropriate to also refer to studies demonstrating the early presence of BPSD in this population, such as the recent study by Dekker et al. (2021) on the BPSD scale.
- Authors should define BPSD in the introduction section, for instance according to the key paper of Finkel in the early 2000s. BPSD in Down syndrome have been described in more detail in the review by Dekker et al. 2015.
- Line 114: irritability and aggressiveness are not for all people with ID early. In Down syndrome, symptoms of anxious behaviour and apathetic behaviour are prevalent (see study mentioned above).
- Line 128: the authors mention issues about caring. Recently, a focus group study was published (DOI: 10.1111/jar.12912) in which caregivers have been asked about such issues.
- Study aim (line 144): the sentence is not entirely clear about whether the study is focused on people with ID + dementia + BPSD or whether people with ID + behavioural changes (possibly BPSD) are also considered.
- Line 149: why are 'other dementia symptoms' also part of the research question. The shift in focus makes the article less powerful.
Methods
- 2.1. Please explain in more detail what clinical vignettes comprise.
- Please do not use too many abbreviations as this makes it difficult to read. Focus groups do not need to be abbreviated. Same goes for LTCF, which I had to Google.
- Could you describe more about the LCTF ?
- Please describe in more detail (2.2/2.3) which professionals were invited and how this was done, e.g. how was the list of eligible FES established? Risk of sampling bias? Lower education/ higher education? Since frontline professionals is not a well-defined term, it remains somewhat vague which group is being addressed.
- 2.4.1. Please provide a short summary of the three clinical vignettes in text, including a few characteristics about the person with ID: age, sex, presence of Down syndrome etc. This helps to interpret the results later on.
- 2.5: could the authors describe why the used phenomenological content analysis instead of thematic analysis? Pros / cons? difference?
- With which population do the FES work in the LTCF? Predominantly mild ID? Predominantly Down syndrome? Predominantly profound ID? Or a mixture? Please describe this, as this is crucial to interpret the results. For instance, recent papers indicate that diagnosing dementia in people with profound ID is more complex than for mild ID, which may also provide different results when asking this to caregivers.
Results
- 3.1/table 2: it is somewhat surprising that nearly 60% of caregivers (frontline workers) has a university degree. Could the authors explain about the care system in Switzerland? In most (European) countries daily caregivers (both medical and non-medical), like nurses, carers, activity supervisors, pedagogical workers, etc. are not university-educated and have received practice-education.
Why do the authors include psychologist etc. in the FES? Are they daily caregivers?
Did the authors study the effect of educational level? It seems to be a binomial division (university / non-university), but isn't this somewhat too simplistic?
- 3.3.: it would be appreciated if table 6 could be moved up, before the different themes are being described.
- Lines 325-340: quote could be shortened.
- Line 357: what is meant with care management tools?
- Section 3.3.3.: a few examples provided are more examples of cognitive decline (forgetfulness) or medical issues (incontinence) than behaviour. it would be advised to stress the behavioural component here as that is the overal research question.
- Could the authors elaborate on whether the results provide any information about whether FES find it more difficult to detect BPSD in mild vs. severe/profound ID? In recent papers, it has been described that for people with profound ID, diagnosing dementia is even more complicated.
Discussion
- Lines 485-487: the authors stress the high prevalence of AD in PWID, but neglect a number of studies that indicate that the prevalence of AD in PWID (Down syndrome excluded) resembles that of the general population. Please provide information/references for both (increased prevalence vs. not increased) as well. For instance: Janicki & Wisnieuwski 1985, Roeden & Zitman, 1994, Aylward 1997, Visser 1997, Zigman 1996 and 2014
- It would be important to mention that not only Down syndrome has an increase risk, but also a few other rare genetic syndromes, like Sanfilippo (strydom 2010), etc.
- Line 499/502: the results largely resemble Clearly/Doody's results indeed. However, a few other recent studies are missing in this section, like the aforementioned focus group study 2021, Iacono 2014, Chapman 2018
- Line 512 - 518: the authors mention two instruments (CAMDEX-DS and DSQIID). Why these two instruments? The authors do not seem aware of a number of other scales and the vast amount of research efforts ongoing in this field. Please see and incorporate the comprehensive review study by Zeilinger et al. 2022. Moreover, in the context of this article, it seems somewhat odd to describe general scales for dementia in people with ID, rather than scales specifically focused on behaviour / BPSD, which are available as well.
- Line 521: could the authors speculate about why professionals with more experience were not more able to recognise BPSD? This is indeed counterintuitive and does not meet what we see in daily practice (various studies, including this one, stress the need for more education in this field)
- 4.2: 'guidelines on diagnosis (...) Down syndrome from age of 35'. How does this recommendation come about from the described results?
- 4.2.: would it not be appropriate to recommend the implementation of a behavioural scale/questionnaires in these long term care facilities to aid the care givers? Not all dementia scales focus comprehensively on behaviour.
Author Response
|
Manuscript ID: disabilities-1911948 |
|
Title: Barriers facing frontline educational staff when supporting older adults presenting with intellectual disabilities and unusual dementia-related behavior: a multi-site, multi-methods study Authors: Karsten Ebbing, Armin Von Gunten, Vincent Guinchat, Dan Georgescu, Taree Bersier, Djamel Moad, Henk Verloo |
Revisions by the authors:
- The first column briefly summarizes each point raised by the editor or reviewer.
- The second column shows our responses to each point.
- The third column gives the location of modifications made to the text or a supplementary file.
|
Nr |
Reviewer 1’s comments This is an important study about how caregivers perceive BPSD in people with ID. Study outcomes are relevant and timely. However, the paper needs revision in a number of major aspects: |
Authors’ responses |
Location in text |
|
1. |
Abstract: - Abbreviation PWID is not common to use. Although it may save words, people in the disability community generally don't appreciate this. Please use either people with ID or people with intellectual disabilities in full. |
As requested by the reviewer, we have replaced PWID with ‘people with ID’ throughout the entire manuscript. We hope that this will increase the paper’s readability and be better received in the disability community |
abstract |
|
2. |
Introduction: |
As requested by both reviewers, we have shortened the introduction and deleted sections of more general information not directly related to the topic. We have added more recent references (1-5), as requested, and restructured the text into paragraphs. |
introduction |
|
3. |
- What is meant with frontline educational staff? This term, as well as its abbreviation FES is very uncommon. How is frontline defined? Don't the same issues present for other staff? Please use 'caregivers' or 'direct support professionals'.
|
We thank the reviewer for these relevant questions.
As mentioned in point 2.2 (lines 176-180), we defined frontline educational staff as: “frontline educational professionals with a Bachelor’s degree, college diploma, or professional education in social work or education, qualifying them to take care of or accompany disabled persons.”
We understand the reviewer’s point. However, as the term ‘FES’ was unusual, the frontline educational staff participating in the study read and approved our manuscript before submission. In fact, they disagreed with the terms ‘direct caregiver’ or ‘support professional’ and wanted to maintain ‘frontline educational staff’ to distinguish their profession from that of nursing staff, as mentioned in the publication by F.K. Sheerin in 2008 (6). This point of view was constructed with regards to the working conditions, salary, and employment contract differences between nurses and educational staff. The authors therefore respected the request by frontline educational staff to be described as such. -We nevertheless adapted the manuscript, using ‘frontline educational staff’ written in full to increase the paper’s readability. We hope that the reviewer finds this acceptable. -We changed “level of impairment” into “level of disability”. - We provided the average life expectancy for people with Down syndrome based on relevant references (7-9). |
Method section (lines 176-180)
Introduction |
|
4 |
- Line 52: the authors state "The degree of impairment—mild, moderate, severe, or profound/complete—is determined by adaptive functioning rather than the intelligence quotient" It might be somewhat confusing, because although adaptive functioning is now regarded more important in defining the level of ID, cognitive (intellectual) functioning needs to be considered as well. IQ is thus no longer leading, but should not be ignored. Please rephrase. Moreover, the authors should realise that in daily practice, IQ is still often being used. |
We considered the reviewer’s comments closely. We propose retaining two definitions: the AAIDD’s and the ICF’s. This enabled us to reword the sentence as: “More recent definitions of ID suggest that the patient’s degree of disability—mild, moderate, severe, or profound/complete—should be determined using adaptive functioning rather than the intelligence quotient. The intelligence quotient is, nevertheless, still often a determining factor in daily practice and in the organization of care and support in long-term care facilities.” |
introduction |
|
5 |
- Page 2: the introduction is rather long and text-book like. Why describe all definitions of intellectual disability? This is not the subject of the article. |
We considered these comments in point 2.
|
introduction |
|
6 |
- Page 2: the authors should use paragraphs. The text is not well-structured and too long. |
We restructured the text into paragraphs, as also requested by reviewer 2, and applied both reviewers’ comments, substantially shortening the text (see point 2). |
introduction |
|
7 |
- Line 77/78: Behavioral changes and functional decline compared to a person’s previous functional abilities are the first signs and symptoms of superimposed dementia [23]. The authors refer to ref. 23 here, which is an interview study with staff. It would be appropriate to also refer to studies demonstrating the early presence of BPSD in this population, such as the recent study by Dekker et al. (2021) on the BPSD scale. - Authors should define BPSD in the introduction section, for instance according to the key paper of Finkel in the early 2000s. BPSD in Down syndrome have been described in more detail in the review by Dekker et al. 2015 |
-We thank the reviewer for this excellent suggestion. We added the reference for Dekker et al. (2021) to support this statement (2). -We considered the reviewer’s suggestion and have added the BPSD definition mentioned within the framework of Down syndrome and also published in Dekker et al. (2015) (10).
We considered the reviewer’s suggestion and have added: “BPSD can be defined as a heterogeneous range of psychological reactions, psychiatric symptoms, and behaviors resulting from the presence of dementia. It is associated with reduced quality of life, increased risk of mortality, accelerated cognitive decline, and earlier institutionalization for patients, as well as severe burdens for caregivers and relatives and increased financial costs (10).” |
Introduction |
|
8 |
- Line 114: irritability and aggressiveness are not for all people with ID early. In Down syndrome, symptoms of anxious behaviour and apathetic behaviour are prevalent (see study mentioned above). |
We considered this suggestion and added the following sentence: “Among older adults with ID, the BPSD are primarily detected following the development of irritability and aggressiveness. Among persons with Down syndrome, these specific symptoms more often lead to a psychiatric consultation than do psychological symptoms like slowing down, apathy, loss of interest, drowsiness, irritation, fear, feeling sad, lack of appetite, and night-time restlessness (10).” |
introduction |
|
9 |
- Line 128: the authors mention issues about caring. Recently, a focus group study was published (DOI: 10.1111/jar.12912) in which caregivers have been asked about such issues. |
We thank the reviewer for this excellent suggestion. We have added the suggested reference (2). |
Introduction |
|
10 |
- Study aim (line 144): the sentence is not entirely clear about whether the study is focused on people with ID + dementia + BPSD or whether people with ID + behavioural changes (possibly BPSD) are also considered. |
We deleted “and other dementia symptoms”. |
introduction |
|
11 |
- Line 149: why are 'other dementia symptoms' also part of the research question. The shift in focus makes the article less powerful |
We deleted the “other dementia symptoms.” |
methods |
|
12 |
Methods - 2.1. Please explain in more detail what clinical vignettes comprise. |
It is usual for psychiatric research to use clinical vignettes to represent real-life care situations. We have added: “A clinical vignette is an abridged report about a patient, summarizing any relevant history, physical examination findings, data from investigations, and treatment information. Vignettes can be used for three main research purposes: to allow actions in context to be explored; to clarify people’s judgements; and to provide less personal and therefore less threatening ways of exploring sensitive topics”. Indeed, using vignettes has been encouraged by several authors, summarized in the scoping review by Tremblay D et al. (2022) (11) |
methods |
|
13 |
- Please do not use too many abbreviations as this makes it difficult to read. Focus groups do not need to be abbreviated. Same goes for LTCF, which I had to Google. |
We thank the reviewer for this comment. This will increase the paper’s readability: we have written LTCF in full as “long-term care facility/facilities.” |
Methods, results discussion |
|
14 |
- Could you describe more about the LCTF ? |
We added what LTCFs’ were to the manuscript: “A long-term care facility can be described as a skilled care facility providing a range of personal and health care services but focusing on medical, educational, and nursing care, and patient safety and educational support in the activities of daily living (12). |
methods |
|
15 |
- Please describe in more detail (2.2/2.3) which professionals were invited and how this was done, e.g. how was the list of eligible FES established? Risk of sampling bias? Lower education/ higher education? Since frontline professionals is not a well-defined term, it remains somewhat vague which group is being addressed. |
To reinforce the description of participant recruitment, we completed the statement as follows: “Based on lists of eligible frontline educational staff presented by participating long-term care facilities, the research team randomly selected 8 participants from each. Each eligible frontline educational worker received information about the study’s objectives and nature, their expected participation, and their provisional appointment with the investigators.” |
methods |
|
16 |
- 2.4.1. Please provide a short summary of the three clinical vignettes in text, including a few characteristics about the person with ID: age, sex, presence of Down syndrome etc. This helps to interpret the results later on. |
We have added the following summary to the manuscript: -In clinical vignette 1, 67-year-old Louise suffers from a mental disability and psychosis and shows increased cognitive disorders and functional decline requiring one-to-one care and support. She progressively refuses to take showers and medication. -In clinical vignette 2, 60-year-old Eugenie is a care home resident with Down syndrome but no superimposed psychiatric disorders. Her behavior suddenly changed in just one day when she fell asleep repeatedly, alternating with periods of crying out and leading to agitated nights. -In clinical vignette 3, 64-year-old Roger has Down syndrome but a high potential for conducting his activities of daily life autonomously. In just a few days he became more apathetic, resistant to care, refused to eat and drink, and showed agitation at night. |
methods |
|
17 |
- 2.5: could the authors describe why the used phenomenological content analysis instead of thematic analysis? Pros / cons? difference? |
The goal of our phenomenological analysis was to describe the essence (or core phenomena) and textures of some conscious psychological experiences. The research team preferred phenomenological analysis because it is the most frequently used qualitative psychiatric clinical research approach used among older adults, and we felt it would best answer our research questions (13). Our phenomenological content approach began with “no a priori assumptions, definitions, or theoretical frameworks” (opening questions in interview guidelines). The focus groups were based on discussions and reflections on participants’ direct perceptions and experiences of the phenomenon being examined (descriptive phenomenology) (14). Indeed, one of the researchers is a medical consultant for the three participating long-term care facilities. A second reason to use phenomenological content analysis was the use of clinical vignettes, with a quantitative content analysis of participants’ answers (qualitative and quantitative approaches). This is why our title uses the term ‘multi-methods study’. |
methods |
|
18 |
- With which population do the FES work in the LTCF? Predominantly mild ID? Predominantly Down syndrome? Predominantly profound ID? Or a mixture? Please describe this, as this is crucial to interpret the results. For instance, recent papers indicate that diagnosing dementia in people with profound ID is more complex than for mild ID, which may also provide different results when asking this to caregivers. |
All the participating long-term care facilities (each hosting more than 300 people with ID) hosted patients with a mixture of mild, moderate, severe, and profound ID (mostly grouped together based on IQ). Special groups existed for people affected by autism spectrum disorders. No special unit existed for people with Down syndrome. Interestingly, all the participating long-term care facilities hosted children as well as “very old people with ID” (min. 2 to max. 88 years old). We did not document the sociodemographic details and disability classifications of the populations in the long-term care facilities because of data protection issues and no clearance to do so from the ethics committee. The ethics committee considered these irrelevant to our research proposal and question. |
results |
|
19 |
Results - 3.1/table 2: it is somewhat surprising that nearly 60% of caregivers (frontline workers) has a university degree. Could the authors explain about the care system in Switzerland? In most (European) countries daily caregivers (both medical and non-medical), like nurses, carers, activity supervisors, pedagogical workers, etc. are not university-educated and have received practice-education. |
The reviewer’s comment is relevant. Two elements explain the study participants’ high level of education. First, the research team received lists from each long-term care facility proposing staff involved in care units for older adults with ID (not staff working with children or young adults). Secondly, to optimize the quality of care and support, long-term care facilities mostly employ trained frontline care staff with bachelor’s or master’s degrees. Thirdly, specialized institutions employ only a small percentage of practice-educated workers, which is why Switzerland’s healthcare costs are among the most expensive in the world (especially in the canton Vaud). |
results |
|
20 |
Why do the authors include psychologist etc. in the FES? Are they daily caregivers? |
Our sample also included some psychologists as frontline educational staff members. This is not exceptional in Switzerland, and many young psychologists wish to experience the daily challenges of people with ID and interprofessional collaborative practice. |
results |
|
21 |
Did the authors study the effect of educational level? It seems to be a binomial division (university / non-university), but isn't this somewhat too simplistic? |
We considered the reviewer’s comments. It is important to mention that we did not explore effects but associations with education levels as an explorative result. We did not aim to explore the differences, and our sample was not calculated to draw out these differences or effects. We agree that the dichotomic university/ non-university analysis eliminates potentially interesting information; however, with a small sample of 24 participants, a more quantitative analysis would not have reached the necessary levels of power. We have added this point to the study’s limitations section: “Quantitative analysis was limited by the small sample, which only enabled basic, dichotomic cross-tabulation statistical tests. |
results |
|
22 |
- 3.3.: it would be appreciated if table 6 could be moved up, before the different themes are being described. |
As the reviewer suggested, we moved Table 6 up the paper. |
results |
|
23 |
- Lines 325-340: quote could be shortened. |
We shortened this quote. |
results |
|
24 |
- Line 357: what is meant with care management tools? |
Most of the long-term care facilities organized their support on the care needs of patients with ID, not on case management (disease management). Frontline educational staff are allocated to people with ID in support-reinforcing “care referent” relationships. Care is also allocated based on the person with ID’s dependency level and complex care needs. |
results |
|
25 |
- Section 3.3.3.: a few examples provided are more examples of cognitive decline (forgetfulness) or medical issues (incontinence) than behaviour. it would be advised to stress the behavioural component here as that is the overal research question. |
We considered the reviewer’s suggestion. We very slightly adapted the relevant quote, but, in our opinion, it would not be correct to change focus group participants’ verbatim quotes. |
Results |
|
26 |
- Could the authors elaborate on whether the results provide any information about whether FES find it more difficult to detect BPSD in mild vs. severe/profound ID? In recent papers, it has been described that for people with profound ID, diagnosing dementia is even more complicated. |
This point is very relevant and could be a step in a new investigation. Nonetheless, we are unable to respond to this question as it was beyond our study’s scope, and our data do not provide us with reliable results. |
Results |
|
27 |
Discussion - Lines 485-487: the authors stress the high prevalence of AD in PWID, but neglect a number of studies that indicate that the prevalence of AD in PWID (Down syndrome excluded) resembles that of the general population. Please provide information/references for both (increased prevalence vs. not increased) as well. For instance: Janicki & Wisnieuwski 1985, Roeden & Zitman, 1994, Aylward 1997, Visser 1997, Zigman 1996 and 2014 |
As requested by the reviewer, we have added additional references (15-17). We added this point to our introduction and have not repeated it in the discussion to avoid redundancy. |
discussion |
|
28 |
- It would be important to mention that not only Down syndrome has an increase risk, but also a few other rare genetic syndromes, like Sanfilippo (strydom 2010), etc. |
We thank the reviewer for this suggestion and have added this information: “Besides people with Down syndrome presenting with an increased risk of dementia, people with other genetic syndromes, such as Sanfilippo, Williams, and Fragile X syndromes, have also been identified as having higher epidemiological rates (18-20). |
discussion |
|
29 |
- Line 499/502: the results largely resemble Clearly/Doody's results indeed. However, a few other recent studies are missing in this section, like the aforementioned focus group study 2021, Iacono 2014, Chapman 2018 |
We thank the reviewer for this suggestion. We have added these references (2, 21). |
discussion |
|
30 |
- Line 512 - 518: the authors mention two instruments (CAMDEX-DS and DSQIID). Why these two instruments? The authors do not seem aware of a number of other scales and the vast amount of research efforts ongoing in this field. Please see and incorporate the comprehensive review study by Zeilinger et al. 2022. Moreover, in the context of this article, it seems somewhat odd to describe general scales for dementia in people with ID, rather than scales specifically focused on behaviour / BPSD, which are available as well. |
We appreciate the reviewer’s suggestions. First, why just two instruments? We limited ourselves to two instruments psychometrically and clinically validated in French, German, and Italian—necessary to cover Switzerland. We have added the existing tools and referenced them with the recent systematic review by Zeillinger et al. (2022) (1). The following sentence has been added to the discussion section: “A recent systematic review by Zeillinger et al. (2022), including 42 eligible publications, summarized the existing ‘Informant-based assessment instruments for dementia in people with intellectual disability.’ About 18 informant-based assessment instruments for screening were found. These authors recommended the Behavioral and Psychological Symptoms of Dementia in Down Syndrome Scale (BPSD-DS), the Cognitive Scale for Down Syndrome (CS-DS), and the Dementia Screening Questionnaire for Individuals with Intellectual Disabilities (DSQIID). For a more thorough dementia assessment, we recommend the Cambridge Examination for Mental Disorders of Older People with Down’s Syndrome and Others with Intellectual Disabilities (CAMDEX-DS) (1)”. |
discussion |
|
31 |
- Line 521: could the authors speculate about why professionals with more experience were not more able to recognise BPSD? This is indeed counterintuitive and does not meet what we see in daily practice (various studies, including this one, stress the need for more education in this field) |
Indeed, we were also surprised to see this result. Our hypothesis was based on the recent phenomenon of ageing older adults with ID. More experienced educational staff have accumulated immense knowledge about the educational paradigm of supporting people with ID but not about dementia semiology. Recent experiences of the superimposition of BPSD on older adults with ID remain in short supply for both experienced and less experienced frontline educational staff. For most of them, this is new, not directly in line with the standard educational paradigm, and sometimes in opposition to evidenced-based dementia care. |
discussion |
|
32 |
- 4.2: 'guidelines on diagnosis (...) Down syndrome from age of 35'. How does this recommendation come about from the described results? |
This recommendation was based more on our authors’ empirical clinical experiences than on our results or recent published research (Karsten Ebbing, Armin Von Gunten, Vincent Guinchat, and Dan Georgescu are consulting psychiatrists in memory clinics and long-term care facilities). |
recommendations |
|
33 |
- 4.2.: would it not be appropriate to recommend the implementation of a behavioural scale/questionnaires in these long term care facilities to aid the care givers? Not all dementia scales focus comprehensively on behaviour. |
We agree with the reviewer, and we have added the additional recommendation that most questionnaires need training to use them. We have added the following additional sentence: “Based on current knowledge and promising psychometric properties, we recommend the use of appropriate, validated behavioral scales or questionnaires to aid frontline educational and nursing staff in documenting unusual dementia-related behavior.” |
recommendations |
Added references:
- Zeilinger EL, Zrnic Novakovic I, Komenda S, Franken F, Sobisch M, Mayer AM, et al. Informant-based assessment instruments for dementia in people with intellectual disability: A systematic review and standardised evaluation. Res Dev Disabil. 2022;121:104148.
- Dekker AD, Wissing MBG, Ulgiati AM, Bijl B, van Gool G, Groen MR, et al. Dementia in people with severe or profound intellectual (and multiple) disabilities: Focus group research into relevance, symptoms and training needs. Journal of Applied Research in Intellectual Disabilities. 2021;34(6):1602-17.
- Dekker AD, Wissing MBG, Ulgiati AM, Bijl B, Gool G, Groen MR, et al. Dementia in people with severe or profound intellectual (and multiple) disabilities: Focus group research into relevance, symptoms and training needs. Journal of Applied Research in Intellectual Disabilities. 2021;34(6):1602-17.
- Dekker AD, Ulgiati AM, Groen H, Boxelaar VA, Sacco S, Falquero S, et al. The Behavioral and Psychological Symptoms of Dementia in Down Syndrome Scale (BPSD-DS II): Optimization and Further Validation. Journal of Alzheimer's Disease. 2021;81:1505-27.
- De Graaf G, Buckley F, Skotko BG. Estimation of the number of people with Down syndrome in Europe. European Journal of Human Genetics. 2021;29(3):402-10.
- Sheerin FK, McConkey R. Frontline care in Irish intellectual disability services: the contribution of nurses and non-nurse care staff. J Intellect Disabil. 2008;12(2):127-41.
- Iulita MF, Garzón Chavez D, Klitgaard Christensen M, Valle Tamayo N, Plana-Ripoll O, Rasmussen SA, et al. Association of Alzheimer Disease With Life Expectancy in People With Down Syndrome. JAMA Network Open. 2022;5(5):e2212910.
- Zhu JL, Hasle H, Correa A, Schendel D, Friedman JM, Olsen J, et al. Survival among people with Down syndrome: a nationwide population-based study in Denmark. Genetics in Medicine. 2013;15(1):64-9.
- O'Leary L, Cooper SA, Hughes‐Mccormack L. Early death and causes of death of people with intellectual disabilities: A systematic review. Journal of Applied Research in Intellectual Disabilities. 2018;31(3):325-42.
- Dekker AD, Strydom A, Coppus AMW, Nizetic D, Vermeiren Y, Naudé PJW, et al. Behavioural and psychological symptoms of dementia in Down syndrome: Early indicators of clinical Alzheimer's disease? Cortex. 2015;73:36-61.
- Tremblay D, Turcotte A, Touati N, Poder TG, Kilpatrick K, Bilodeau K, et al. Development and use of research vignettes to collect qualitative data from healthcare professionals: a scoping review. BMJ open. 2022;12(1):e057095.
- Egan C, Mulcahy H, Naughton C. Transitioning to long-term care for older adults with intellectual disabilities: A concept analysis. J Intellect Disabil. 2021;0(0):17446295211041839.
- Larsen RR, Maschião LF, Piedade VL, Messas G, Hastings J. More phenomenology in psychiatry? Applied ontology as a method towards integration. The Lancet Psychiatry. 2022;9(9):751-8.
- Gutland C. Husserlian Phenomenology as a Kind of Introspection. Frontiers in Psychology. 2018;9.
- Janicki MP, Wisniewski HM. Aging and Developmental Disabilities—Issues and Approaches. Baltimore: Oxford Brookes University; 1985.
- Janicki MP, Dalton AJ. Prevalence of dementia and impact on intellectual disability services. Ment Retard. 2000;38(3):276-88.
- Liao P, Vajdic C, Trollor J, Reppermund S. Prevalence and incidence of physical health conditions in people with intellectual disability – a systematic review. PLOS ONE. 2021;16(8):e0256294.
- Strydom A, Chan T, King M, Hassiotis A, Livingston G. Incidence of dementia in older adults with intellectual disabilities. Res Dev Disabil. 2013;34(6):1881-5.
- Evans E, Bhardwaj A, Brodaty H, Sachdev P, Draper B, Trollor JN. Dementia in people with intellectual disability: insights and challenges in epidemiological research with an at-risk population. Int Rev Psychiatry. 2013;25(6):755-63.
- Sauna-Aho O, Bjelogrlic-Laakso N, Siren A, Arvio M. Signs indicating dementia in Down, Williams and Fragile X syndromes. Molecular Genetics & Genomic Medicine. 2018;6(5):855-60.
- Iacono T, Bigby C, Carling-Jenkins R, Torr J. Taking each day as it comes: staff experiences of supporting people with Down syndrome and Alzheimer's disease in group homes. J Intellect Disabil Res. 2014;58(6):521-33.

Reviewer 2 Report
Thank you for the opportunity to review your manuscript regarding the barriers facing frontline educational staff supporting older adults with intellectual disabilities and dementia related behaviors. It is likely a more common situation for this population of adults than is documented. Your study could provide a starting point for so much more regarding more reliable ways to diagnose the occurrence and support the adults, their families and their support care staff. I do have some recommendations to support the readability (including some points of clarity) of the manuscript.
1. The opening paragraph is very long. I would suggest breaking it into several paragraphs- potentially line 64, line 74, and line 88 could be the start of new paragraphs. I also recommend editing line 544 to avoid starting a paragraph with "This".
2. I would suggest a careful read to use person first language (e.g., line 170) as we want to respectfully recognize the person first and their disability second.
3. I am confused as to what exactly the participants did with the vignettes in section 2.4.1 and the subsequent table of results. I don't understand the "unclear" options. Does that mean the participants were unclear what to do or that their responses were unclear for coding? The items seem to flip between what the participants ratings (item 1) and participant behavior (item 2). Who made the judgements about items 3 and 4 in the table? I am not sure how it could be "unclear" if the participant mention barriers or propose solutions for each vignette. More information about the vignettes, what participants did, and what the ratings in Table 3 mean is necessary.
4. Tables 2 and 3 should be reversed in order as the authors write about Table 3 in line 219 but Table 2 is line 247.
5. I would delete line 242 as it is redundant with line 179. I would also either use line 272 or use the first part of line 255 but both are not needed.
6. I am unsure who is being referred to by "them" in line 265.
7. Which test was used for education level in Table 4?
8. Line 287 seems to be a mistake or incomplete text or a header.
9. The language in Table 6 does not match the headings in Section 3.3. They should be the same.
10. In the Discussion, research is cited differently than the rest of the paper (e.g., 485). The method for citations should be consistent.
11. I think it is misleading to proport this student could help with prevention of BPSD (lines 496, 562). The discussion section should discuss the potential of the study within the research purpose- perceived barriers to detecting BPSD and supporting PWID presenting those behaviors AND identify ways to support the care givers to provide best practice. If prevention of BPSD is part of one of those purposes, the connection was not made clear.
12. For the recommendation starting line 616, is middle age too late as line 102 stated that individuals with Down syndrome could develop dementia as early as age 35?
Author Response
|
Manuscript ID: disabilities-1911948 |
|
Title: Barriers facing frontline educational staff when supporting older adults presenting with intellectual disabilities and unusual dementia-related behavior: a multi-site, multi-methods study Authors: Karsten Ebbing, Armin Von Gunten, Vincent Guinchat, Dan Georgescu, Taree Bersier, Djamel Moad, Henk Verloo |
Revisions by the authors:
- The first column briefly summarizes each point raised by the editor or reviewer.
- The second column shows our responses to each point.
- The third column gives the location of modifications made to the text or a supplementary file.
Reviewer 2
|
34 |
1. The opening paragraph is very long. I would suggest breaking it into several paragraphs- potentially line 64, line 74, and line 88 could be the start of new paragraphs. I also recommend editing line 544 to avoid starting a paragraph with "This". |
We thank the reviewer for their suggestions. We have shortened and restructured the introduction section. We changed line 544 to: “Our findings about the connection between experience and knowledge …” |
introduction |
|
35 |
2. I would suggest a careful read to use person first language (e.g., line 170) as we want to respectfully recognize the person first and their disability second. |
We examined the use of person-first language throughout the manuscript, applying this suggestion where possible (as also suggested by Reviewer 1 in point 1). |
manuscript |
|
36 |
3. I am confused as to what exactly the participants did with the vignettes in section 2.4.1 and the subsequent table of results. I don't understand the "unclear" options. Does that mean the participants were unclear what to do or that their responses were unclear for coding? The items seem to flip between what the participants ratings (item 1) and participant behavior (item 2). Who made the judgements about items 3 and 4 in the table? I am not sure how it could be "unclear" if the participant mention barriers or propose solutions for each vignette. More information about the vignettes, what participants did, and what the ratings in Table 3 mean is necessary. |
We considered the reviewer’s comments. We added the following comments to clarify Table 3’s results: “All the participants received the three clinical vignettes and their accompanying open questions, as mentioned in Suppl. File 1. The clinical vignette questionnaires were completed before the focus groups. The participants were asked to answer to the best of their knowledge and using their experiences in daily practice. Two authors (KE and HV) analyzed and coded the clinical vignettes based on a grid especially constructed for the study. If the written answer did not clearly mention that the behavior observed was related to the BPSD, then the evaluators considered this to be “unclear”. Evaluations were made independently by two researchers (KE and HV), with disagreements resolved by discussion.” |
methods |
|
37 |
4. Tables 2 and 3 should be reversed in order as the authors write about Table 3 in line 219 but Table 2 is line 247. |
As suggested by the reviewer, we inversed Tables 2 and 3. |
results |
|
38 |
5. I would delete line 242 as it is redundant with line 179. I would also either use line 272 or use the first part of line 255 but both are not needed. |
As suggested by the reviewer, we deleted lines 242 and 272. |
results |
|
39 |
6. I am unsure who is being referred to by "them" in line 265. |
We thank the reviewer for pointing this out. We replaced “them” with “participating frontline educational staff”. |
results |
|
40 |
7. Which test was used for education level in Table 4? |
We added the statistical test for education in Table. |
results |
|
41 |
8. Line 287 seems to be a mistake or incomplete text or a header. |
We thank the reviewer for this remark, and we corrected line 287. |
results |
|
42 |
9. The language in Table 6 does not match the headings in Section 3.3. They should be the same. |
We matched the headings in Table 6 with Section 3.3. |
results |
|
43 |
10. In the Discussion, research is cited differently than the rest of the paper (e.g., 485). The method for citations should be consistent. |
We made the citation method consistent throughout the manuscript. |
discussion |
|
44 |
11. I think it is misleading to proport this student could help with prevention of BPSD (lines 496, 562). The discussion section should discuss the potential of the study within the research purpose- perceived barriers to detecting BPSD and supporting PWID presenting those behaviors AND identify ways to support the care givers to provide best practice. If prevention of BPSD is part of one of those purposes, the connection was not made clear. |
We considered these comments and deleted lines 496 and 562 to avoid any misleading or unsupported statements in our results. |
discussion |
|
45 |
12. For the recommendation starting line 616, is middle age too late as line 102 stated that individuals with Down syndrome could develop dementia as early as age 35? |
We considered the reviewer’s comments and have slighty adapted the sentence in line 616. Line 102: “Although approximately 15%–40% of people with Down syndrome over the age of 35 present with a clinical picture of AD. Because the onset of dementia is so early, the average age of people with Down syndrome and AD is estimated to be 51.3 years old ”
|
discussion |
Added references:

Round 2
Reviewer 1 Report
I thank the authors for considering the input. I believe that their changes have contributed strongly to improvement of the manuscript.
To my opinion, the term frontline educational staff remains somewhat odd. In their answer, it seems as if the staff determined their own functional description, rather than adhering to terminology in similar scientific publications.
Author Response
Dear Editor,
I understand the remaining comments of reviewer 1. I consulted the long-term care facilities to see if they agreed to modify the "frontline education staff" into "direct support professionals". If the Editor agrees with this change, I will proceed to revise the manuscript and present an adapted revision.
I hope that this proposition is acceptable. I thank you for your reply and acceptance.
With my best regards
Henk Verloo